# Analyses and Correlation of Pathologic and Ocular Cutaneous Changes in Murine Graft versus Host Disease

**DOI:** 10.3390/ijms23010184

**Published:** 2021-12-24

**Authors:** Robert B. Levy, Hazem M. Mousa, Casey O. Lightbourn, Eric J. Shiuey, David Latoni, Stephanie Duffort, Ryan Flynn, Jing Du, Henry Barreras, Michael Zaiken, Katelyn Paz, Bruce R. Blazar, Victor L. Perez

**Affiliations:** 1Department of Microbiology and Immunology, University of Miami Miller School of Medicine, Miami, FL 33101, USA; col8@med.miami.edu (C.O.L.); sduffort@med.miami.edu (S.D.); hbarreras@med.miami.edu (H.B.); 2School of Medicine, Duke University, Durham, NC 27708, USA; hm146@duke.edu (H.M.M.); ericshiuey@gmail.com (E.J.S.); david.latoni_morales@tufts.edu (D.L.); 3Department of Pediatrics, Division of Blood & Marrow Transplant & Cellular Therapy, University of Minnesota, Minneapolis, MN 55455, USA; rflynn85@gmail.com (R.F.); jing.du@northwestern.edu (J.D.); zaike002@umn.edu (M.Z.); katelyngoodman2@gmail.com (K.P.); blaza001@umn.edu (B.R.B.)

**Keywords:** ocular GVHD, cutaneous GVHD, correlation, pathology, ocular immune response, inflammation

## Abstract

Graft versus host disease (GVHD) is initiated by donor allo-reactive T cells activated against recipient antigens. Chronic GVHD (cGVHD) is characterized by immune responses that may resemble autoimmune features present in the scleroderma and Sjogren’s syndrome. Unfortunately, ocular involvement occurs in approximately 60–90% of patients with cGVHD following allo-hematopoietic stem cell transplants (aHSCT). Ocular GVHD (oGVHD) may affect vision due to ocular adnexa damage leading to dry eye and keratopathy. Several other compartments including the skin are major targets of GVHD effector pathways. Using mouse aHSCT models, the objective was to characterize cGVHD associated alterations in the eye and skin to assess for correlations between these two organs. The examination of multiple models of MHC-matched and MHC-mismatched aHSCT identified a correlation between ocular and cutaneous involvement accompanying cGVHD. Studies detected a “positive” correlation, i.e., when cGVHD-induced ocular alterations were observed, cutaneous compartment alterations were also observed. When no or minimal ocular signs were detected, no or minimal skin changes were observed. In total, these findings suggest underlying cGVHD-inducing pathological immune mechanisms may be shared between the eye and skin. Based on the present observations, we posit that when skin involvement is present in aHSCT patients with cGVHD, the evaluation of the ocular surface by an ophthalmologist could potentially be of value.

## 1. Introduction

Allogeneic hematopoietic stem cell transplantation (aHSCT) has become the standard of care for many hematopoietic malignancies and genetic disorders. Despite advances in achieving allo-engraftment, graft versus host disease (GVHD) remains a major obstacle to the more widespread usage of this cellular therapy [1,2,3,4,5]. While advances in GVHD prophylaxis and treatment have resulted in higher survival rates post-HSCT, patient quality of life is compromised in part as a result of ocular complications, such as dry eye, leading to loss of visual acuity and, in some cases, blindness [6,7,8,9,10]. Unfortunately, it is estimated that ocular GVHD is present in 60–90% of patients with chronic GVHD manifestation [11]. Moreover, dermatologic manifestations are also common and significantly affect patients by causing skin, often scleroderma-like, lesions and promoting infection, thereby limiting the patient’s ability to function on a daily basis [12,13].

Notably, in murine experimental aHSCT models, we and others have identified ophthalmic manifestations that may correlate with dermatologic changes during the course of systemic chronic GVHD [14,15,16,17,18,19]. To date, we are not aware of reports that have evaluated and compared ocular changes and cutaneous alterations in controlled experiments using established mouse models of GVHD. Accordingly, we posited that these two GVHD target tissues may share similar immune mechanisms leading to correlative damage and clinical manifestations. To rigorously test this hypothesis, in this study we examined multiple well-defined experimental GVHD mouse models. This included experiments to examine for correlations in animals undergoing therapeutic treatment using post-transplant cyclophosphamide (PTCy) as a single agent for GVHD prophylaxis, which others and our labs have demonstrated to ameliorate GVHD [20,21]. 

To address these questions, allogeneic HSCT was performed using mouse models of MHC-matched and MHC-mismatched donor/recipient combinations. In MHC-matched aHSCT recipients, skin changes represented by the thickening of the keratinized epithelium and infiltration of the dermis by mononuclear inflammatory cells were accompanied by ocular manifestations represented by lid margin edema and corneal superficial punctate keratopathy in recipients that had increased clinical scores. A correlation between skin and ocular manifestations was also observed using the MHC mismatched (H2b → H2k and H2b → H2d) models. Interestingly, one of the MHC-mismatched transplant combinations (H2b → H2d) exhibited both skin and ocular changes. In contrast, the other combination (H2b → H2k) did not, i.e., neither skin nor ocular damage was observed.

In total, using well-established multiple pre-clinical mouse aHSCT models, we found that there were consistent and statistically significant correlations identified between the presence or absence of ophthalmic and cutaneous manifestations of GVHD. Therefore, similar to patients following aHSCT where clinical complications associated with GVHD vary, our work suggests that in mice that developed GVHD the same correlations between ocular GVHD and scleroderma may exist, and this should be taken into consideration as a potential biomarker to guide systemic and local treatment of this disease. 

## 2. Materials and Methods

### 2.1. Animals

All animal studies were conducted according to protocols approved by the University of Miami, Minnesota, and Duke University’s Animal Care and Use Committees, and in accordance with the ARVO Statement for the Use of Animals in Ophthalmic and Vision Research. C57Bl/6 (B6; H2b), C3H.SW (H2b), LP/J (H2b), B10.BR(H2k), BALB/c (H2d), and B10.D2 (H2d) (NCI, Taconic, or The Jackson Laboratory) and maintained at university facilities. All mice used in experiments were 8–10 weeks old, exhibited no obvious ocular surface and eyelid disease at baseline, and were fed with a standard caloric diet for their age. The animals were maintained in pathogen-free conditions at the respective University animal facilities and routinely monitored prior to all procedures until the end of the experiment.

### 2.2. Hematopoietic Stem Cell Transplantation (HSCT)

Major MHC-mismatched HSCT models and use of cyclophosphamide post-transplant: B6->BALB/c (H2d) received ablative conditioning with a single dose of 8.5 Gy total body irradiation (TBI, X-ray and Cesium) 1 day prior to transplant. Bone marrow (BM) cells were obtained from femurs ± tibias, and vertebrae from sex-matched B6-CD45.1 (H2b; Thy1.2) donor animals. A single-cell suspension of BM cells was prepared by flushing bones with a 21-gauge needle filtering through a 100-μm nylon mesh. T-cell depletion (TCD) of donor marrow cells was achieved via complement-mediated lysis using anti-T-cell-specific antibody HO-13-4 (hybridoma supernatant, mouse anti-Thy1.2 IgM; ATCC), anti-CD4 (clone 72.4) mAb, anti-CD8 (clone H02.2) mAb (initially provided by Dr. Bruce Blazar, University of Minnesota, Minneapolis, MN, USA), and rabbit complement (Cedarlane Laboratories, Burlington, ON, Canada). The marrow cells were incubated at 37 °C for 45 min, washed twice in RPMI, and resuspended for hematopoietic cell transplant. Marrow TCD was routinely >99%. Donor T cells were prepared from spleens obtained from B6-FoxP3rfp animals. Donor cells were stained for T cells (anti-CD4, clone RM4-5; anti-CD8, clone 53-6-7) and adjusted to 1.1 × 10^6^ T cells per mouse prior to mixing with BM. Recipient mice were transplanted (day 0) with TCD BM (5 × 10^6^) and 1.1 × 10^6^ T cells IV in a 0.2 mL volume via tail vein injection. 

In some experiments, as indicated, cyclophosphamide was administered on days 3 and 4 (50 mg/kg i.p. per injection) post-transplant (i.e., PTCy) for single agent GVHD prophylaxis as previously described by us and others [20,21]. Additionally, in some experiments, transplanted mice also received TL1A-Ig (50 μg/injection × 4) and low dose IL-2 (10,000 units rhIL-2/injection × 2), following PTCy using a protocol we have previously reported to stimulate the TNFRSF25 and CD25 receptors in vivo [22,23].

B6→B10.BR recipients were conditioned with cyclophosphamide on days 3 and 2 (120 mg/kg per day intraperitoneally). On day 1, recipients received TBI by X-ray (8.3 Gy). B6 donor BM was TCD (T cell depleted) with anti-Thy1.2 monoclonal antibody (mAb), followed by a rabbit complement. Splenic T cells were purified by negative selection using anti-CD19, anti-B220, anti-CD11b, anti-CD11c, anti-TCRg/d, anti-NK1.1, and anti-TER119 antibodies and Stemcell Rapidsphere magnetic beads (StemCell Technologies, Vancouver, BC, Canada). On day 0, recipients received 10 × 10^6^ TCD BM cells ± 70–75,000 T cells [24].

### 2.3. MHC-Matched HSCT Models

B10.D2 BM (10 × 10^6^) and T cells (2.7 × 10^6^; 2:1 CD4:CD8 ratio), purified as above, were given to Balb/c recipients conditioned with 7.5Gy TBI by X-ray (day 1). The mice were monitored daily and assessed for clinical score and skin score as described previously [25,26]. B6 (H2b) → C3H.SW (H2b) recipients were conditioned with 10.5 Gy (n = 8) using a Cs137 source 3–4 h prior to transplantation. Cell suspensions containing donor B6 bone marrow and T cells were prepared as described above for the B6→BALB/c transplants and adjusted in a serum-free medium to a concentration of 4.6 × 10^6^/mL for intravenous (0.5 mL) injection of 2.3 × 10^6^ T cells/mouse. LP/J (H2b)→B6 (H2b) recipients were conditioned with 8.5 Gy TBI from a Cs137 source on day 0. The mice were injected intravenously (IV) with TCD-BM (1 × 10^7^) with or without splenic T cells (8 × 10^5^). 

### 2.4. Clinical Evaluation of Systemic and Ocular Graft vs. Host Disease

Systemic and cutaneous GVHD was assessed by monitoring recipients for changes in total body weight, clinical signs, and overall survival as previously described [22,26]. The clinical signs of GVHD were recorded for individual mice. Recipients at the University of Miami and Duke were scored on a scale from 0 to 2 for 6 clinical parameters. We adapted our scoring for clinical parameters—a majority of which was originally obtained from Cooke et al. [27]—to a “scale from 0 to 2 for 6 clinical parameters including diarrhea”, as described by Perez et al. [15]. The systemic parameters included were: (1) weight loss; (2) diarrhea; (3) fur texture; (4) posture; (5) alopecia; and (6) mobility. Additionally, for skin scoring performed at the University of Minnesota, cutaneous changes were scored as previously described [25]. 

Ocular GVHD: To monitor and score ocular clinical pathological changes, at each time-point of analysis, individual animals were evaluated and graded from 0–4 for the clinical parameters (1) eyelid and (2) corneal involvement, modified from Perez et al. [15]. 

### 2.5. Histology and Analyses of Corneal and Cutaneous Pathological Changes

To assess and compare the histologic findings between the GVHD and BMO groups, the microscopic examination of histology slides was performed and scaled photographs were obtained.

### 2.6. Details of Staining and Sectioning

Eyes and skin punch biopsy samples from the different experimental and control mice were collected and frozen into tissue blocks. Frozen tissue blocks containing the eyes were sectioned using the cryotome at the desired thickness (10–12 μm). The sections were then collected by mounting on slides. Then, the slides were left to air dry for 3–5 min after which they were stained with 0.1% Mayer’s hematoxylin (Sigma: MHS-16) for 5–10 min in a 50 mL conical tube. After rinsing with running ddH2O for 5 min, the sections were dipped in 0.5% eosin 12 times and then sequentially dipped in H2O (10 times), 50% ethanol (10 times), and 70% ethanol (10 times). The slides were then equilibrated in 95% ethanol for 30 s and 100% ethanol for 1 min, before being dipped in xylene multiple times. Finally, a coverslip was placed with Cytoseal XYL (Stephens Scientific).

### 2.7. Disease Comparison and Statistical Analyses

To assess for clinical disease association between ocular and skin GVHD, MHC-matched (B6 → C3H.SW) allogeneic transplant of donor bone marrow only (n = 5) and bone marrow + T cells (n = 5) was carried out. Each mouse was scored 3 times a week for ocular and systemic disease, as described previously (Section 2.4). The disease parameters were plotted as mean +/− SD to identify trends and a descriptive analysis was used to identify associations.

The histologic findings were evaluated and compared between the GVHD and BM groups; the microscopic examination of histology slides was performed and photographs were obtained. For the ocular scores, gross changes identified included (1) corneal epithelial thickening, (2) stromal thinning, and (3) vacuolization of the corneal epithelium. For the skin score, parameters identified included (1) dermal thickening, (2) epidermal thickening, and (3) loss of hair follicles. These parameters were measured and quantified across the different eye tissue harvested from transplanted animals. A Mann–Whitney non-parametric t-test was used for statistical analysis to compare the median of both BM + T disease and BMO control groups and a *p*-value of < 0.05 was considered significant. 

A composite ocular and a composite skin histology disease score were obtained to allow testing for correlation between the different disease findings (ocular and skin). Both control (BM only) and experimental (BM + T cells) groups were grouped, and the median and interquartile range were calculated. For each of the 3 ocular and 3 skin disease parameters, the cut-off for the first quartile (Q1), the median, and the cutoff for the third quartile (Q3) were then used as the scoring cutoffs to determine the score of disease for each category. For each of the 6 parameters, scores ranged from 0–3 with a score of 0 for values within the 1st quartile, a score of 1 for values within the 2nd quartile, a score of 2 for values within the 3rd quartile, and a score of 3 for values within the 4th quartile. For each of the ocular parameters and skin parameters, the scores from each of the parameters were then added to achieve a composite score for each mouse ranging from 0–9, 1 for ocular disease and 1 for skin disease. Spearman’s rank correlation coefficient was used to determine the correlation between the calculated composite ocular and composite skin disease scores. Only mice with both ocular and skin histologic measures were included which led to a sample size of n = 8. Spearman’s r coefficient was calculated and a *p*-value < 0.05 (denoted by an asterisk “*”) was used to determine the significance of correlation.

## 3. Results

### 3.1. Kinetic Analysis of Changes in the Eye and Skin following Pre-Clinical MHC-Matched Allogeneic Hematopoietic Stem Cell Transplants 

To investigate for a potential relationship between ocular and skin tissue involvement (Figure 1A) in GVHD, we began with the clinical evaluation of mice undergoing allogeneic hematopoietic stem cell transplants (HSCT) using the scoring system for skin (Methods) and our ocular clinical scoring system (Figure 1B) [15,16]. Five distinct donor-recipient strain combinations were used throughout these studies to assess MHC-matched (n = 3) and mismatched (n = 2) transplant outcomes (Table 1). Initially, an MHC-matched transplant was performed using B6 (H2b) donors and C3H.SW (H2b) recipient mice, which result in ocular GVHD [14,15]. Notably, systemic clinical signs of GVHD are readily apparent 2–3 weeks post-HSCT in animals receiving both donor BM and splenic T cells compared to recipients of BM only (Figure 2A). Following the development of systemic acute GVHD at approximately 3 weeks post-HSCT, changes were observed in both the skin and eye. Skin changes are reflected by an increased clinical score (ruffling, alopecia, and scabbing) and histological changes (thickening and inflammatory infiltrate). Eye findings included eyelid edema and the corneal clinical epithelial haze and keratopathy with histologic changes demonstrating epithelial thickening and infiltration (Figure 2B–D). Notably, the kinetics of eyelid involvement paralleled that observed in recipient skin (Figure 2B vs. Figure 2D). Interestingly, corneal clinical changes were observed later, 5–7 weeks post-HSCT.

We then examined additional MHC-matched donor-recipient strain combinations. First, B10.D2 (H2d) → BALB/c (H2d) transplants were investigated. Eyelid swelling as well as corneal keratopathy and skin changes (i.e., ruffling, alopecia, and scabbing) were again apparent in the recipients of donor BM and T cells (Figure 3A).

The histological examination demonstrated epithelial corneal thickening (blue arrow) and corneal stromal thinning (black arrow) in animals undergoing GVHD, which was not observed in control BM only transplanted mice (Figure 3B). the histological examination of the skin (Figure 3B: lower panels) showed epidermal layer thickening, dermal thickening with extensive collage nation infiltrating the underlying fat layer, and loss of hair follicles in recipients of T cells undergoing GVHD, but not in recipients of BM only controls. To quantitate these changes, we developed a scoring criterion to assess epithelia thickening and corneal stromal thinning as described (Methods).

The results indicated significant increases in corneal as well as skin epithelial thickening in mice undergoing GVHD (Figure 3C, upper panels) as well as significant increases in their total ocular and skin scores (Figure 3C, lower panels). Although recipients of BM and T cells exhibited other histological trends, such as decreased corneal stromal thickness and higher corneal epithelial vacuolization, differences compared to recipients of BM alone did not attain statistical significance (Appendix A). MHC-matched transplants were also performed utilizing a third donor-recipient strain combination, i.e., LPJ (H2b) → B6 (H2b) recipients. In the manner of the other MHC-matched models, ocular and cutaneous clinical changes were clearly observed after 1 month involving lid edema, conjunctival swelling, and keratopathy as well as fur changes and skin scabbing, respectively (Appendix A).

### 3.2. Analysis of Ocular and Cutaneous Changes following MHC-Mismatched Allogeneic Hematopoietic Stem Cell Transplants

MHC-mismatched donor/recipient aHSCT was performed using a model reported to reflect chronic GVHD as indicated by changes in a wide variety of recipient compartments, including the lung and mucosal and lymphoid tissues [24]. C57BL/6 (H2b) bone marrow +/− 6 × 10^4^ T cells were administered into B10.BR (H2k) recipients (Figure 4). Two independent experiments (n = 5/group) were performed, and animals were examined at ~1 month (28 days) and ~2 months (56 days) post-HSCT. Mild clinical changes were typically detected in recipients by 2 months post-HSCT were present in the skin (ruffling) and eyes (mild lid edema) of recipients of bone marrow plus T cells (Figure 4A). Moreover, histological analyses of the cornea and skin indicated minimal epithelial thickening and quantification of scoring confirmed there was not statistical significance versus control recipients (Figure 4C). In total, these findings again indicated a positive correlation between the skin and eye compartments based on significant (Figure 3) or mild (Figure 4) involvement in these preclinical transplant models (Figure 5).

Next, to further test this correlation, transplants were performed with BALB/c (H2d) recipients of donor B6 (H2b) BM +/− T cells using an acute model of GVHD (Figure 6). The examination of eye and skin in these mice revealed that, following allogeneic BM + T cell transplant, mice with no or low (less than or equal to 1.0) clinical eyelid scores had no skin score as well (Figure 6A). We have previously utilized PTCy post-transplant (PTCy) for GVHD prophylaxis in pre-clinical HSCT models [20,23]. In this study, PTCy was administered for GVHD prophylaxis in this MHC-mismatched B6 → BALB/c transplant. Similar to the non-PTCy treated animals, the assessment of eye and skin in these recipients revealed that, following allogeneic BM + T cell transplant, mice with no or low clinical eyelid scores had no skin score (closed symbols, Figure 6B). Notably, some animals receiving PTCy did develop GVHD and, in all but one of these mice, eye scores were significantly greater and skin scores were also elevated (open symbols, Figure 6B). One mouse treated with PTCy did exhibit significant eye scores without noticeable skin changes (open upward triangle, Figure 6B). Notably, we also performed transplants in this model in which animals received T cells from donor B6 mice and were then administered TNFRSF25 and CD25 agonists, which we have previously reported stimulate the Treg compartment [22,28] (Figure 6C). None of these recipients exhibited strong eye or skin scores (all scores ≤1, Figure 6C). These data also support the notion that eyelid compared to skin scoring is a more sensitive indicator of ongoing GVHD and, importantly, a positive correlation was observed between changes detected in the eye and skin compartments [15].

## 4. Discussion

There have been several drugs recently approved for use in patients with GVHD, including ibrutinib, ruxolitinib, and most recently, belumosudil [29,30,31,32,33]. Ruxolitinib has led to a diminishment of treatment-related morbidity, reduction of GVHD, and prolonged patient survival [30,31,32]. Nonetheless, ocular GVHD occurs in the majority of patients with chronic GVHD following aHSCT, resulting in damage to ocular tissues, dry eye disease, and keratopathy [6,7,8,9,10]. In this context, it is interesting that cutaneous manifestations are the most common—and frequently the presenting—clinical sign of GVHD and involve rash, erythroderma, and blisters, and scleroderma-like lesions [34,35]. Notably, the use of murine experimental aHSCT models has observed ophthalmic manifestations, such that analysis of potential correlation with dermatologic changes can be investigated [14,15,16]. It is well established that different pre-clinical mouse HSCT models as observed in patients after aHSCT exhibit varying levels of GVHD severity in different target compartments, including the gastrointestinal tract, lungs, as well as the eye and skin [36,37,38,39]. Since both the ocular and skin compartments contain an important epithelial cell layer directly exposed to environmental stresses, we considered the notion that GVHD might similarly affect both of these tissues and employed pre-clinical HSCT to examine if any correlation could be identified.

We first examined multiple models of experimental allogeneic MHC-matched HSCT that represent the most common aHSCT performed in our patients to assess ocular and skin changes and identified a consistent correlation between ocular and cutaneous involvement accompanying GVHD. Additionally, we exploited our group’s experience across multiple institutions and performed experimental MHC-mismatched aHSCT to determine if this correlation also occurred. We detected a “positive” correlation, i.e., when GVHD-induced ocular changes were detected in these complete MHC-mismatched recipients, changes in the cutaneous compartment were typically observed. In contrast, when low or no ocular involvement was detected, minimal skin changes were observed. Secondly, we observed that examination of the MHC-mismatched H2b (B6) into H2d (BALB/c) model demonstrated a correlation regardless of the extent of eye involvement, hence when there was a low eye involvement, there was a low skin involvement, and when high eye involvement was detected, there was a correspondingly high skin score. Notably, to more rigorously analyze clinical changes across the skin and cornea, histological samples from the bone marrow only controls and that of the diseased GVHD mice (both MHC-matched and mismatched models) were assessed, comparing the skin disease severity index with that of the ocular severity index (Figure 5). When the total ocular disease score index for each mouse was plotted against the total skin disease index, a significant correlation between skin and eye disease was identified. Overall, these findings suggest underlying GVHD-inducing pathological immune/inflammatory mechanisms may be shared between the eye and skin tissues.

Lastly, to directly test the correlation between ocular and cutaneous correlation, we performed interventional experiments by manipulating the T-cell compartment in vivo to assess cutaneous and ocular responses. Interestingly, when PTCy was employed for GVHD prophylaxis, we observed a predominant diminution of systemic GVHD (data not shown) in most animals, as anticipated [20,21]. However, GVHD was observed in some animals with both ocular and cutaneous clinical manifestations of GVHD. We do not know and cannot predict these outcomes, but posit that some animals did develop GVHD resulting from incomplete deletion and/or regulation of donor anti-host alloreactive T cells specific for MHA and/or non-MHC encoded minor transplantation antigens. Our prior work demonstrated that Treg cells are required for effective PTCy prophylaxis of GVHD [21]. Hence, when Treg cells were not present, the emergence of GVHD after cyclophosphamide presence at D.3 and 4 strongly suggests some donor T cells with anti-host reactivity do persist and need to be regulated, i.e., suppressed to prevent GVHD [21]. Notably, in mice in which we administered agonists to expand Treg cells post-transplant and after cyclophosphamide treatment, very mild systemic GVHD was detected; however, a correlation was observed between ocular (mild) and skin (mild) involvement. It should be noted that two of the animals treated with PTCy did not show a correlation between the eye and skin (Figure 6B). Interestingly, in these animals, the eyelid showed a significant score without skin involvement, suggesting that the eyelid may be a more sensitive indicator of GVHD, as we previously reported [15]. Overall, we speculate that diminishment of cGVHD following PTCy treatment in patients will predominantly reflect the skin [34,35] and ocular correlations we identified, and therefore, in recipients with a diminished cutaneous GVHD, a diminished oGVHD will be observed. Moreover, although additional studies are required, we can speculate the potential generalizability of these findings onto other prophylactic regimens in GVHD (i.e., anti-thymocyte globulin, G-CSF, etc.) at ameliorating GVHD resulting from shared immune mechanisms across different tissues [40].

To our knowledge, the studies we are reporting represent the initial observations using experimental allo-hematopoietic stem cell transplant models selectively examining two target tissues affected by GVHD, the eye and skin. While the sample sizes for this first study are modest, we remain confident in the results and the associations observed. Moreover, we anticipate the work will stimulate additional studies that will continue to address the hypothesis that similar immune mechanisms are responsible for GVHD induced ocular and skin manifestations." In total, using multiple independent and well-established pre-clinical mouse aHSCT models, regardless of the donor and recipient strains employed, we consistently observed a correlation between the ophthalmic and cutaneous compartments in recipients undergoing GVHD. Accordingly, when ophthalmic involvement was identified, skin alterations also were present in the recipients (i.e., “positive correlation”) and, in contrast, when ophthalmic changes did not occur, skin involvement was also not identified. Therefore, not unlike clinical aHSCT in which complications associated with GVHD vary, the findings in this study suggest that, in patients with GVHD, a similar correlation may exist in the clinical setting. Therefore, when skin involvement is present in aHSCT recipients with GVHD, the evaluation of the ocular surface by an ophthalmologist could be of potential value.

## Figures and Tables

**Figure 1 ijms-23-00184-f001:**
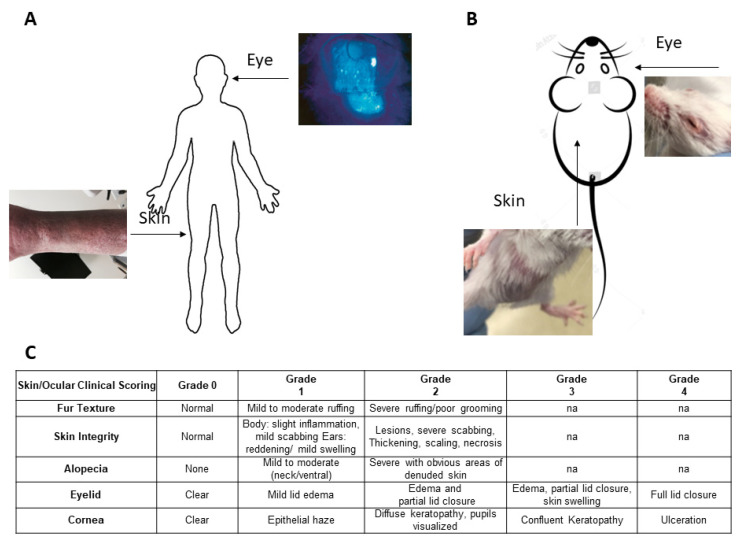
Schematic and clinical scoring features of ocular and cutaneous involvement in graft vs. host disease. (**A**) Clinical photos from patient with GVHD with corneal keratopathy (as demonstrated with fluorescein staining) and sclerodermal skin changes characterized by the thickening and induration of the cutaneous epithelium. (**B**) Schematic of ocular changes characterized by lid swelling and closure and skin pathology as demonstrated by alopecia and decreased skin integrity and fur texture in pre-clinical HSCT models. (**C**) Skin and ocular scoring system parameters used in the GVHD models were adapted from Perez et al. [15] and Cooke et al. [27].

**Figure 2 ijms-23-00184-f002:**
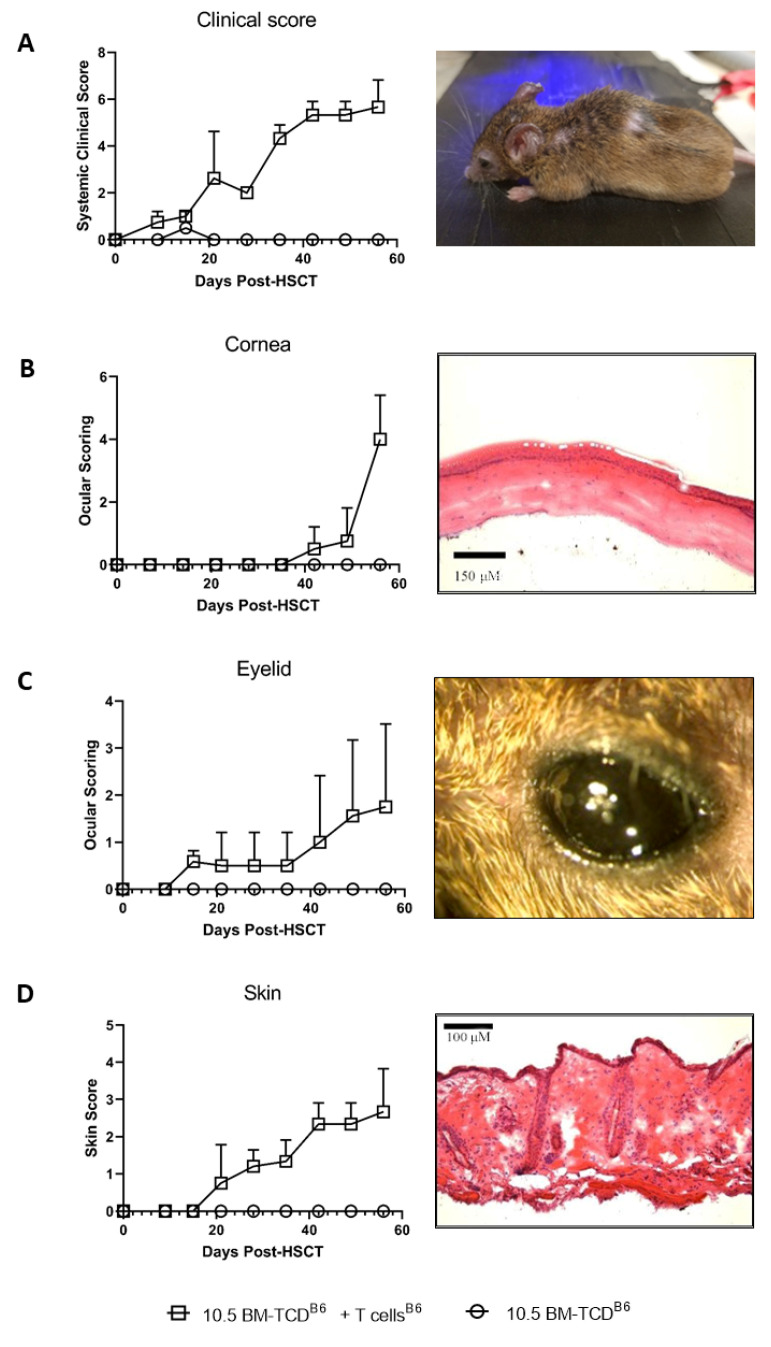
Correlation of the development of ocular and skin GHVD in MHC-matched HSCT recipients. Correlation between cutaneous and ocular tissue involvement (plotted as mean and SD) following MHC-matched (B6 → C3H.SW) allogeneic transplant of donor bone marrow (T-cell depleted—TCD-BM) +/− T cells. Mice (n = 5/group) were transplanted with donor B6 TCD-BM alone did not exhibit detectable changes in the skin or ocular compartment. Recipients of B6 TCD-BM + T cells initially exhibited a systemic clinical score (**A**) and then cutaneous GVHD (**B**) and ocular GVHD involving the cornea and lids (**C**,**D**). The data in this figure is representative of an individual experiment. This donor/recipient combination has been routinely transplanted in our laboratory and systemic and ocular GVHD changes were reported in several published studies [14,15,21].

**Figure 3 ijms-23-00184-f003:**
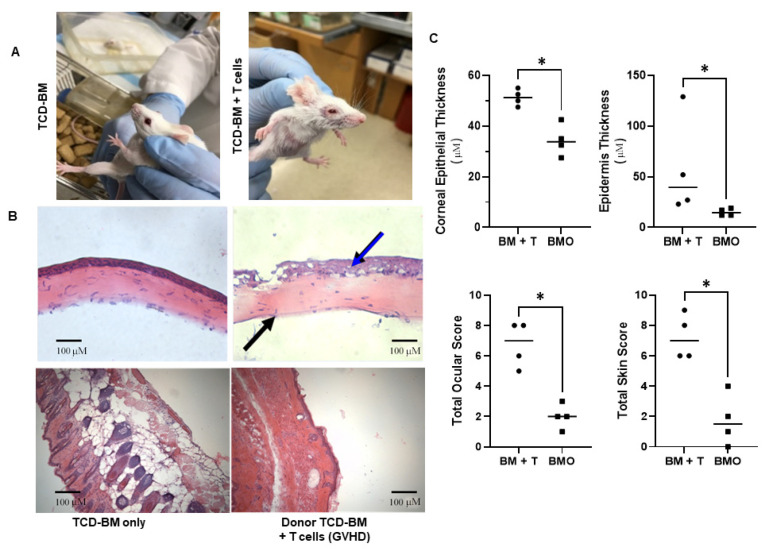
Development of ocular and skin GVHD following HSCT in an MHC-matched donor/recipient strain combination. Transplants were performed using donor B10.D2 (H2d) bone marrow +/− spleen cells containing T cells into BALB/c (H2d) recipients. (**A**) Clinical examination of animals approximately 5-weeks post-HSCT revealed that mice receiving donor T cells exhibited ruffled fur, alopecia, and scabbing in the ears. In the ocular compartment, edema with eyelid swelling was observed. This analysis is based on an n = 4/group (animals meeting the inclusion criteria, i.e., animals for which we had both skin and eye histology allowing analysis) for BMO and BM + T cells. ((**B**), upper panels) The histological examination of the cornea demonstrated epithelial thickening (blue arrow) and corneal stromal thinning (black arrow) in animals undergoing GVHD but not in control transplanted mice. ((**B**), lower panels) The histological examination of the skin comparing bone-marrow only control (B, lower left panel) with GVHD samples ((**B**), lower right panel) revealed epidermal layer thickening, dermal thickening with extensive collage nation infiltrating the underlying fat layer, and loss of hair follicles. (**C**) Significant differences were observed between control and GVHD mice in the epidermal thickening in the ocular and skin compartments ((**C**): upper panels). Analysis of the total histological ocular and skin scores ((**C**): lower panels). BM + T cells (n = 4/group) and BMO (n = 4/group). * denotes *p* value < 0.05.

**Figure 4 ijms-23-00184-f004:**
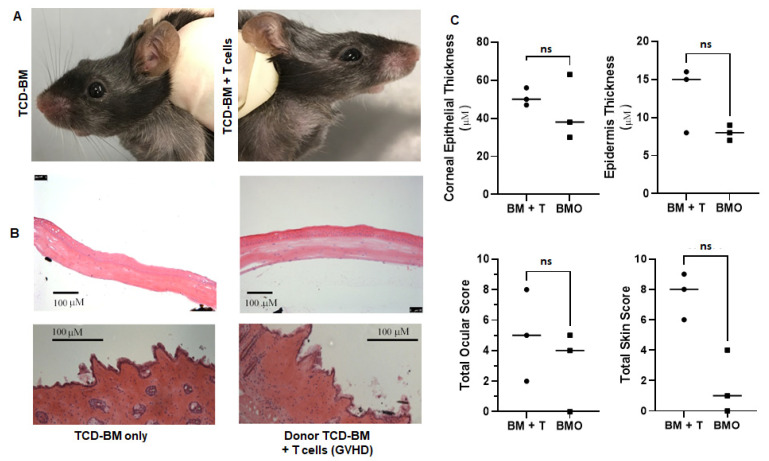
Development of ocular and skin GVHD following HSCT in an MHC-mismatched donor/recipient strain combination. Transplants were performed using donor B6 (H2b) bone marrow +/− spleen cells containing T cells into B10.BR (H2k) recipients. (**A**) Clinical examination of animals approximately 8-weeks post-HSCT revealed marginal clinical with virtually no edema and eyelid swelling in the ocular compartment in mice receiving BM only or BM + T cells. Only mice with both ocular and skin histologic samples were included. This led to n = 4/group in both BM + T cells and BMO groups. (**B**) The histological examination of the cornea did not identify epithelial thickening in the cornea or skin or corneal stromal thinning. Data presented represent individual B10.BR animals at Day 56. (**C**) The analysis of representative mice found the total histological ocular and skin scores were not significantly different between recipients of BM only vs. BM + T cells. ns denotes *p* values that were >0.05.

**Figure 5 ijms-23-00184-f005:**
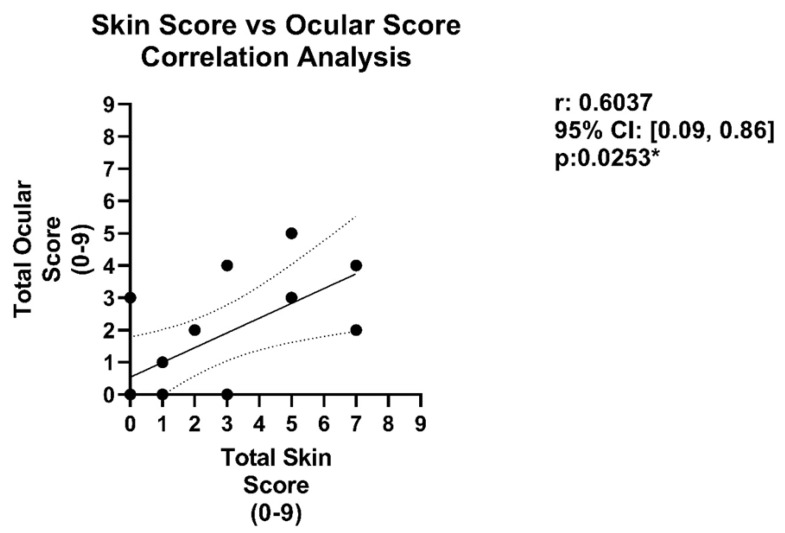
Correlation analysis comparing skin disease severity index with that of the ocular severity index at 8-weeks post-HSCT. Summary of results and comparisons of the skin disease variables and skin severity scores for the MHC-matched (B10.D2 → BALB/c), MHC-mismatched (B6 → B10.BR) GVHD models, and controls (bone marrow only). This included 4 mice from each GVHD model (n = 8 total) When the total ocular disease score index for each mouse was plotted against the total skin disease index, a significant correlation between skin and eye disease was identified: coefficient of 0.6037 (*p*-value: 0.0253).

**Figure 6 ijms-23-00184-f006:**
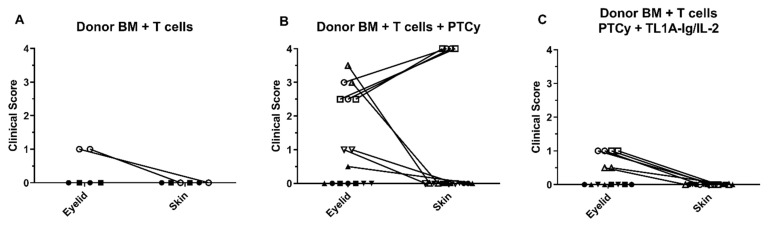
Analyses of ocular and skin GVHD in untreated and recipients treated early following HSCT in MHC-mismatched recipients. Transplants were performed with donor B6 TCD-BM + T cells into 8.5 gy conditioned BALB/c recipients. (**A**) Two levels of severity in animals were identified in mice surviving 6 weeks: one exhibiting low eyelid involvement and virtually no skin involvement and the other no eyelid or skin involvement. (**B**) Mice transplanted as described were treated on days 3 and 4 post-transplants with cyclophosphamide (“PTCy”, 50 mgs/kg). The levels of severity identified in (**A**) were again observed. A third level group exhibited significant eyelid and skin involvement. (**C**) Mice were transplanted and treated with PTCy as described in (**B**) and received treatment initiated at day 5 with anti-TNFRSF25 and anti-CD25 agonists to stimulate the Treg compartment. (See Methods Section). Again, the same levels of severity as identified in panel A were identified in mice 6 weeks post-transplant, one exhibiting low eyelid involvement and virtually no skin involvement and the other no eyelid or skin involvement. The cumulative ocular eyelid score was determined by adding the scores (see Figure 1) of each eye from a single mouse. Each line represents the eye and skin score of the same individual animal. Lines of points on the x-axis cannot be visualized. The open and closed symbols represent animals with a cumulative eyelid score of ≥1 and <1, respectively.

**Table 1 ijms-23-00184-t001:** Transplants performed to assess/examine the relationship between skin and eye involvement post-allo HSCT.

Donor Strain	Recipient Strain	Genetic Disparity
B6 (H2b)	C3H.SW (H2b)	MHC Matched ^1^
LP/J (H2b)	B6 (H2b)	MHC Matched
B10.D2 (H2d)	BALB/c (H2d)	MHC Matched
B6 (H2b)	B10.BR (H2k)	MHC Mismatched ^2^
B6 (H2b)	BALB/c (H2d)	MHC Mismatched

^1^ Major histocompatibility matched, minor histocompatibility mismatched transplants. ^2^ Major histocompatibility mismatched, minor histocompatibility mismatched transplants.

## Data Availability

To share data throughout the research community, and all can benefit from the tools, reagents and models generated at the institutions, pending third parties rights, the institutions will transfer materials to outside researchers under a Material Transfer Agreements (MTAs) generated and monitored by the University of Miami and Duke Technology Transfer Offices. Such MTAs will be made with no more restrictive terms than the Simple Letter Agreement (SLA) to non-profit institutions or the Uniform Biological Material Transfer Agreement (UMBTA) to for-profit ones. This publication also functions as a means to share knowledge obtained from our studies.

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
