# Peer review of "Analyses and Correlation of Pathologic and Ocular Cutaneous Changes in Murine Graft versus Host Disease"

_ijms, 2021, doi:10.3390/ijms23010184_

Round 1
Reviewer 1 Report
In this manuscript, Levy et. al characterize ocular and dermal chronic GVHD across multiple mouse models. Chronic GVHD can manifest in any organ system, and its onset and pathogenesis can be difficult to predict and analyze. Ocular GVHD presents in a majority of patients who develop chronic symptomology after allogeneic HSCT, although dermal GVHD is the most common presentation. Given the structural similarities between skin and eye surfaces, the authors’ aim was to determine whether ocular and cutaneous manifestations of chronic GVHD correlated with each other across multiple pre-clinical mouse models of HSCT. These models encompassed a broad swath of GVHD models, using both fully MHC-matched (miMHC-mismatched; analogous to a standard clinical usage of allogeneic HSCT) and fully MHC-mismatched murine models of GVHD. The use of post-transplant cyclophosphamide in one model further adds to the clinical relevance of the work.
The manuscript is well-written, with good grammar and sentence structure. However, figures (including the supplement) are quite blurry throughout and a solution needs to be found to make them clearer, either on the authorial or journal side. How statistics were calculated for each figure was often unstated, and I would like to see the addition of a statistics section in the methods to clarify what was done, especially in regards to the non-standard p value of <0.1 denoting significance rather than the more common p < 0.05. While this is not wrong per se, the authors’ adoption of this in the absence of an explanatory statistics section is concerning. Additional statistics throughout the paper were unclear (figure 3, all p values identical; figure 4, all p values rounded off) or absent (figure 2). Additionally, animal n’s did not always match up to the numbers shown in figures, and no explanation is given for this. Lastly, some data seems to be overinterpreted, especially when considering the preclinical nature of the models being used.
Overall, the manuscript and science are good and will contribute to the field, but much needs to be done to clarify the authors’ actions in the preparation of the document such that the science therein is clear and repeatable.
Line 21: These conclusions seem to me to be a slight overinterpretation of the data. The “negative” correlation is not necessarily present, but could be instead due to incomplete penetrance of the model, IE not every mouse develops clinical GVHD. This statement, used repeatedly, should either be addressed more thoroughly such that the possibility of incomplete penetrance is discussed, or softened.
Lines 24-25 and 376-377: This is preclinical work, and the wording of “necessary” or “required” ophthalmologist examination is premature. While an examination is certainly not contraindicated based on these data, I do not believe they are of sufficient strength to “require” a change in clinical practice of this magnitude in the absence of additional human data.
Line 37: Dermal GVHD is more than just scleroderma. Expand on this slightly.
Lines 58-60: This wording is a bit jumbled, should maybe be broken up into two sentences to clarify.
Line 80: The sentence ends prematurely.
Lines 96, 97, 111, 113, 119, 120, 122, possibly others: Change these to 10^6 or 106. You have the latter usage at the end of line 122, but it should be standardized throughout.
Line 103: This is merely referred to as “cyclophosphamide” everywhere else in the manuscript except here.
Lines 123, 124, 134, 138: The colon from the end of line 123 appears to have wandered down to line 124. The italics on line 124 should also be reverted to normal text. Lines 134 and 138 are missing colons at the end of their titles.
Line 128: This reference from Cooke et. al does not include diarrhea or alopecia measurements. While the parameters for these are clarified further down in Figure 1 and seem scientifically appropriate, it should be noted that these are either previously unpublished scoring criteria, or from a reference other than Cooke et. al.
Line 130: Reference number 25 here has gotten merged with the text.
Lines 136 and 151: “Scaled photographs” are mentioned but nowhere in the manuscript does any photo have a visible scale bar.
Line 145: “H20” should be “H2O”
Line 156: This is addressed in the introduction to the review as well, but I would like to see some justification for the choice of p < 0.1 as significant. A statistics section should be added to the methods to explain what tests were run and how significance was calculated for each assay.
Figure 1: Figures are blurry and the chart is slightly compressed on the x-axis (see where “Skin/Ocular Clinical scoring” is cut off). I would recommend moving Figure 1B up and right to be parallel with 1A, then expanding the chart. The chart also needs a “Figure 1C” designation.
Figure 2: Figures are blurry and should be enlarged, especially the microscopy insert photos. I would like to see the location of figures 2B and 2C switched to improve readability. Why is only one representative graph shown? This figure should show all 5 mice with error bars indicating SD or SEM. If not all mice developed disease, that would be indicative of incomplete penetrance in this model, as mentioned in my comment on Line 21.
Line 197: There is a missing arrow or hyphen between B6 and C3H
Line 198: “group” should be written out.
Figure 3: The whole figure is quite blurry. While the overall phenotype is evident based on the pictures shown, the pictures are too small and too blurry to see the details of what is shown. The pictures should be increased in size and resolution, and scale bars should be added per lines 136 and 151. Figure 3C is also blurry, and all the p values are identical, and no data is included as to how these p values were calculated. In addition, why are only 4 mice shown in the figure when there were 6 mice per group?
Line 218: “in” should be removed
Figure 4: In terms of image quality, it needs improvement as noted in my comment on Figure 3. Here, all p values seem to be rounded off, and no information is included as to how they were calculated. In figure 4C, again, you have 11 and 8 mice per group, but why are only a few mice analyzed? These n’s should be increased or justified.
Figure 5: What timepoint were the scores taken at for these mice? Was this the maximum score for each mouse, was it taken at a standard timepoint, or how was this calculated? Why are there only two seemingly disparate (one MHC-mismatched, one MHC-matched) transplants shown here, and why not include others. Even if the LP/J model was not scored, the C3H model was scored and should be included in this.
Line 319: Should be amended to “pre-clinical HSCT models”
Line 324: “Groups” should be “group’s”
Line 332: Should be “when there was low eye involvement”
Line 371: Again, I’m skeptical as to whether this is a negative correlation or just a lack of active disease in these models.
Line 376: Remove “particular”
Lastly, not all author names are spelled correctly.
Reviewer 2 Report
In this review article, Levy et al. present data on various types of allogeneic transplantation in mice (donor types, T-cells and immune-modulatory therapy) and the observed clinical outcomes in respect to ocular and skin GVHD. The article is well written and the data are clearly presented. However, some aspects must be addressed before the article can be accepted.
It is well known that experiencing acute GVHD significantly increase the risk of cGVHD, and that presence of cGVHD in one organ predicts involvement in other organs. Since skin and ocular cGVHD are one of the two most frequently observed forms of cGVHD, you need high numbers and a well-designed experiment to prove/ disprove the co-occurrence of these manifestation
Furthermore, GVHD, and especially cGVHD, is difficult to examine due to the heterogeneity of clinical presentation. In the current article data show either a high degree of acute/chronic GVHD and a correlation between skin and ocular GVHD. The problem with mice models and GVHD is that you need a nuanced model and large numbers to clearly demonstrate an association or no association. It is well known that if you observe a high degree of acute GVHD you would also observe a high degree of chronic ocular and skin in individuals experiencing GVHD and vice versa. The authors state that “The cornea and skin indicated minimal epithelial thickening and quantification of scoring confirmed there was no statistical significance versus control recipients” However, the numbers of experiments used in this comparison is very low and actually no clear conclusion can be drawn. This all comes down to the level of difference that you regard as relevant and the sufficient number that you need to confidently rule out that an observed difference/or no-difference is present.
Furhter comments:
- The authors should also include a short discussion on how other types than post-transplant cyclophosmaid as GVHD prophylaxis might influence ocular GVHD, e.g ATG and G-CSF primed bone marrow.
- Line 10 “against recipient allo-transplantation antigens”. This is incorrect, the donor derived lymphocytes is directed against host antigens, not allo antigens.
- Line 10/11 Chronic GVHD is not characterized by autoimmune features. cGVHD is an allo-immune response that resembles features of different autoimmune disorders. Please rephrase
- I am not sure that 60+% of patients experience ocular GVHD, this seems like an overestimate. Ocular GVHD is sometimes hard to distinguish from lacrimal damage due to previous chemotherapy. In our hospital doing 100+ transplant/ year, cGVHD is seen in 40% while extended cGVHD is seen in 20% after the implication of ATG as GVHD prophylaxis
- Line 14 “Ocular GVHD (oGVHD) affects vision due to ocular adnexa damage leading to dry eye 1”. Please rephrase… Ocular GVHD may affect…
- Line 51, please state that these transplantations were done in mice as experimental modes. It is clear from the rest of the article (line 862) that this was done in mice, however, since this is the introduction I recommend that the authors make this clear a bit earlier.
- Line 105 please use the term cyclophosphamide and not cytoxan
Round 2
Reviewer 1 Report
The manuscript is much improved. Increased methodological clarity and improvements to figure resolution have been added to the manuscript. Thank you for taking the time to address my concerns.
The only comments are minor: Table 1 should maybe be on its own page (this can be addressed with the journal editorial office). Secondly, Figure 4C, top left graph, has both "*" and "NS".
